# Improving the Pharmacological Properties of Ciclopirox for Its Use in Congenital Erythropoietic Porphyria

**DOI:** 10.3390/jpm11060485

**Published:** 2021-05-28

**Authors:** Ganeko Bernardo-Seisdedos, Jorge M. Charco, Itxaso SanJuan, Sandra García-Martínez, Pedro Urquiza, Hasier Eraña, Joaquín Castilla, Oscar Millet

**Affiliations:** 1ATLAS Molecular Pharma S. L. Parque Tecnológico de Vizcaya, Ed. 800, 48160 Derio, Spain; gbernardo.atlas@ciocbiogune.es (G.B.-S.); jmoreno.atlas@cicbiogune.es (J.M.C.); gmartine.sandra@gmail.com (S.G.-M.); herana.atlas@cicbiogune.es (H.E.); jcastilla@cicbiogune.es (J.C.); 2CIC bioGUNE, BRTA, Parque Tecnológico de Vizcaya, Ed. 800, 48160 Derio, Spain; isanjuan@cicbiogune.es (I.S.); urquiza.ortiz@gmail.com (P.U.); 3IKERBASQUE, Basque Foundation for Science, Prion Research Lab, 48011 Bilbao, Spain

**Keywords:** ciclopirox, pharmacological chaperones, porphyria, drug discovery, protein stability

## Abstract

Congenital erythropoietic porphyria (CEP), also known as Günther’s disease, results from a deficient activity in the fourth enzyme, uroporphyrinogen III synthase (UROIIIS), of the heme pathway. Ciclopirox (CPX) is an off-label drug, topically prescribed as an antifungal. It has been recently shown that it also acts as a pharmacological chaperone in CEP, presenting a specific activity in deleterious mutations in UROIIIS. Despite CPX is active at subtoxic concentrations, acute gastrointestinal (GI) toxicity was found due to the precipitation in the stomach of the active compound and subsequent accumulation in the intestine. To increase its systemic availability, we carried out pharmacokinetic (PK) and pharmacodynamic (PD) studies using alternative formulations for CPX. Such strategy effectively suppressed GI toxicity in WT mice and in a mouse model of the CEP disease (*UROIIIS*^P248Q/P248Q^). In terms of activity, phosphorylation of CPX yielded good results in CEP cellular models but showed limited activity when administered to the CEP mouse model. These results highlight the need of a proper formulation for pharmacological chaperones used in the treatment of rare diseases.

## 1. Introduction

Shortcomings in the pharmacological properties constitute one of the main sources in the failure of clinical trials for drugs that have proven active in preclinical studies [1]. To be safe, a drug must be completely eliminated from the body, ideally not long after the activity’s window timeframe [2]. To that end, the drug catabolism (pharmacokinetics, PK) has to be fine-tuned with the biochemical and physiological effect of the drug (pharmacodynamics, PD). Improved PK can be achieved by the modification of the active principle, yet this may be at the expense of the drug’s PD. Alternatively, a complex formulation may help PK optimization without jeopardizing the active principle’s efficacy, but this strategy may be of limited applicability. In this study, we have explored all these strategies over ciclopirox, a topical antifungal that has been repositioned as a potential drug for the treatment of congenital erythropoietic porphyria (CEP), acting as a pharmacological chaperone [3].

Porphyrias, inborn errors of heme biosynthesis, are metabolic disorders, each resulting from the deficiency of a specific enzyme in the heme biosynthetic pathway [4,5]. This group of diseases includes CEP (ICD-10 #E80.0; MIM#263700), also known as Günther’s disease [6,7,8]. CEP is autosomal recessive and results from a markedly deficient activity of the uroporphyrinogen III synthase (UROIIIS; EC 4.2.1.75) that leads to the accumulation of type I porphyrins, specifically uroporphyrin I (URO I) and coproporphyrin I (COPRO I) [9]. The accumulation of these porphyrins leads to the specific symptoms of this disease, such as hemolysis, severe anemia, splenomegaly, and disfiguring phototoxic cutaneous lesions [4].

CEP is a mutilating and one of the most severe porphyrias, and it currently has no curative treatment other than bone marrow transplant, an approach that is not devoid of specific risks including infections derived from immunosuppression, toxicity problems derived from chemotherapy, transplant rejection, and, eventually, premature death of the patient [7]. Palliative care includes avoidance of sun exposure, skin care, and avoiding mechanical trauma [10]. 

We recently demonstrated the medical plausibility of ciclopirox (CPX) for the treatment of CEP, acting as a pharmacological chaperone targeting uroporphyrinogen III synthase [3]. Pharmacological chaperones function by directly binding a folded or partially folded protein to stabilize it and allow completion of the folding process to yield a functional protein [11,12,13]. In turn, CPX is a topical treatment of cutaneous fungal infections and is believed to act as a fungicidal agent by chelating polyvalent metal cations such as Fe^3+^ and Al^3+^, resulting in the inhibition of peroxide degradation [14]. CPX binding to UROIIIS stabilizes its structure and reduces its unfolding and degradation with time [3]. Therefore, CPX restores the protein levels of UROIIIS and its activity. The effect of CPX on the activity of UROIIIS was evidenced in cell-based and murine models of CEP [15]. CPX caused a significant decrease in the levels of the toxic porphyrins, particularly URO I and COPRO I, in liver, red blood cells, and urine. Furthermore, it reduced splenomegaly, an indirect measure of reduction in circulating porphyrins [3]. Altogether and considering that CEP is an ultra-rare disease, CPX was granted an orphan drug designation for the treatment of CEP by the FDA (DRU-2018-6297, May 2018) and the European Medicines Agency (EMA/OD/186/17, January 2018).

The PK of CPX was initially described during its development as an antifungal agent [16]. Several studies were performed with oral administration, for instance, at a dose of 1 mg ciclopirox-^14^C-olamine/kg to rats, or in doses between 10 and 15 mg ciclopirox-^14^C-olamine/kg to dogs. In such experiments, the preparation of CPX-olamine was either encapsulated as a crystallizate in hard gelatin capsules (for dogs) or dissolved in polyethylene glycol 400 (for rats) [17]. The results indicate that the compound is quickly eliminated in urine (3–6 h). Despite this fast turnover, our own experience and the literature [16] indicate that CPX administered orally to mice results in gastrointestinal toxicity (GI toxicity). Figure 1A shows the effect of CPX accumulation in the gut, with macroscopic bowel inflammation. This results are also consistent with a study in rats where ciclopirox-olamine was administered to rats for 4 weeks at doses up to 300 mg/kg found gastric irritation and chronic gastritis [18]. This acute toxicity evidences the need for further development of the drug. Herein, we have first developed an NMR-based method for the monitorization of CPX and related compounds in animal models. This method allowed investigation of the activity and catabolism of a CPX prodrug, intended to circumvent the toxicity problems [19]. The prodrug is able to solve the observed GI toxicity without altering the activity of the active principle, but at the expense of the pharmacokinetic profile.

## 2. Materials and Methods

### 2.1. Compounds and Cell Lines

The compound ciclopirox (CPX, 6-cyclohexyl-1-hydroxy-4-methyl-2(1H)-pyridinone) was purchased from Santa Cruz Biotechnology (sc-204688), and phosphorylated ciclopirox (CPX*pom*, O-phosphoryl-methylen-6-cyclohexyl-1-hydroxy-4-metyl-2(1H)-Pyridinone) was synthetized by Charnwood Molecular Ltd. PD of CPX was studied in HEK CRISPR Cas UROIIIS-C73R and UROIIIS-P248Q cell lines [20]. Briefly, 60 µM dose of CPX and phosphorylated CPX was administrated for 30 min and 1, 2, and 4 h of exposition. Then, the cells were counted and harvested. Samples were treated by the same protocol for the porphyrin extraction. *Mouse experiments*. For the PK experiments and for each *bolus* administration method 10 WT ICR (CD-1^®^) outbred mice (Envigo) were used. Serum was collected after the administration at 1, 2, 4, 6 (7), 12, and 24 h. At 48 (36) h, the mice were sacrificed. Triplicates were collected for each data point. Urine samples were collected at the same interval. In the PD experiments, the concentration of porphyrins was measured from circulating blood. Blood samples were extracted from the submandibular vein every week, and mice were weighed before each extraction. Mice were treated with oral CPX by gavage to evaluate CPX dose efficiency. In such experiment, CPX was administered every 24 h for 6 consecutive weeks, and the mice were weighed every week. All work performed with animals was approved by the competent authority (Diputación Foral de Bizkaia) following European and Spanish directives. The CIC bioGUNE Animal Facility is accredited by AAALAC Intl.

### 2.2. Sample Preparation for NMR Spectroscopy

An aliquot of 200 µL of murine serum was placed in a 1.5 mL Eppendorf. Afterwards, 1.3 mL of MeOH:H_2_O (ratio 2:1) was added. The mixture was gently shaken until final homogenization. Samples were centrifuged for 30 min at 20,000× *g*. Supernatant was transferred to a new tube and dried in a SpeedVac, which was resuspended in 480 µL of DMSO-d6 with 1.66 µM of DSS as an internal reference (sodium trimethylsilylpropanesulfonate) and placed into a 5 mm NMR tube. Urine samples were collected in an Eppendorf tube and immediately frozen until further use. Samples were thawed on ice and diluted in D_2_O till it makes a volume of 450 µL. Further, 1 mM of sodium azide was added for sample conservation and 100 µM of DSS was added as reference and placed into a 5 mm NMR tube.

### 2.3. NMR Spectroscopy

NMR data were collected at on an 800-MHz Bruker Avance III spectrometer equipped with a cryoprobe and on a 600-MHz Bruker Avance III US2 spectrometer. For each sample, a 1D ^1^H noesygppr1d spectrum was collected (Bruker, size of fid 69228, ds 16, ns 1024, d1 3 s, experiment time 55 min). Data analysis was done using the TopSpin 3.5 software (Bruker BioSpin GmbH). Free induction decays were multiplied by an exponential function equivalent to 0.3 Hz line-broadening before applying Fourier transform. All transformed spectra were corrected for phase and baseline distortions and referenced to the DSS singlet at 0 ppm. Chemical shifts for CPX*pom*, CPX*hm*, CPX, and CPX*glu* were predicted in silico [21] and assigned by spike. Metabolic quantification was carried out at peaks’ integral against the added internal reference compound. In case of signal overlap, peak deconvolution (command LDCON) was done to assign corresponding peak areas. Final CPX serum/urine concentration was obtained considering peaks’ spin system and sample dilution performed during sample preparation.

### 2.4. Porphyrin Extraction

Murine blood samples were obtained from the submandibular vein and collected in ethylenediaminetetraacetic acid tubes; samples were aliquoted and stored in a freezer at −80 °C. For porphyrin extraction, 300 µL of 6 M hydrochloric acid was added to the cell samples and 200 µL to the blood samples, then sonicated for 3 cycles at 25” each and incubated at 37 °C for 30 min at 450 rpm. The samples were then centrifugated for 10 min at 10,000× *g*. The pellet was removed, and the supernatant was transferred to a centrifuge tube filter cellulose acetate membrane, pore size 0.22 μm, and centrifugated for 10 min at 4000× *g*. The samples were then analyzed by HPLC.

### 2.5. HPLC Analysis

Porphyrins from cell lines were separated by HPLC analysis on a ODS Hypersil C18 column (5 µm, 3 mm × 200 mm; Thermo Scientific, Waltham, MA, US) in a HPLC chromatograph (Shimadzu, Long Beach, CA, US). Porphyrins were separated with a 60 min gradient elution and a two-component mobile phase consisting of ammonium acetate (1 M, pH 5.16, solvent A) and 100% acetonitrile (solvent B) at a flow rate of 1 mL/min. All analyses were performed at 20 °C, and porphyrins were detected by fluorescence with an excitation wavelength 405 nm and emission wavelength 610 nm.

### 2.6. Pharmacokinetic Analysis

Non-compartmental serum and urine pharmacokinetic parameters for CPX*pom* and CPX were determined using SimBiology module in MATLAB. Values from IV administration of the drugs were fitted to an exponential function (a·e^(b*t)^) assuming maximum concentration at t = 0. For oral administration, values were fitted to a rational polynomial ((a1*t + a2)/t^2^ + b1·t + b2)) assuming 0 at t = 0.

## 3. Results 

### 3.1. CPX Characterization in Biofluids (Serum and Urine) by NMR Spectroscopy

We first explored the use of NMR for the identification/quantification of CPX and its derivatives in biofluids, in the context of PK studies. NMR spectroscopy is well suited for the xenobiotic characterization of urine and serum as it is quantifiable, reproducible, non-selective, and non-destructive [22,23].

In the liver and other tissues [24], xenobiotic compounds are metabolized by direct compound modifications (phase I biotransformations: oxidation, reduction, hydrolysis, etc.) and by conjugation reactions (phase II biotransformations). For CPX, UDP-glucuronosyltransferase transfers the glucuronic acid component of uridine diphosphate glucuronic acid to CPX to produce the glucuronidated derivative (CPX*glu*), which is much more soluble and it is quickly excreted in the urine [25]. Actually, this occurs mostly during the first passage to the liver and most of the CPX circulating in serum corresponds to CPX*glu*. Figure 2 shows the assignment of the signals in the serum spectrum (of mouse), where two doublets at 6.23 and 6.03 ppm are characteristic for CPX (6.5 and 6.41 ppm in urine). These signals belong to the cyclohexyl moiety of the compound and, therefore, they remain unperturbed upon derivatization (i.e., they account for the total circulating CPX, CPX*tot*). The intensity of the signal can be easily converted into absolute concentration by normalization with respect to a reference compound (see Materials and Methods). In turn, the doublet signal at 4.86 ppm was assigned to the glucuronic moiety of CPX*glu*, and it can be used to directly quantify this species (5.18 ppm in urine). Of note, we can only use the right-half of the doublet as the left-half is overlapped with other signals from the serum matrix.

The free amount of CPX (CPX*free*) is estimated by the difference between CPX*tot* and CPX*glu*:CPX*free* = CPX*tot* − CPX*glu*.(1)

Herein, it is important to mention that the determination of CPX*free* is not exactly accurate (i.e., around 50% error) because it relies on the difference of two concentrations that are one order of magnitude higher. In any case, our results demonstrate that the NMR spectrum analysis can provide a quantitative analysis of CPX catabolism (serum, Appendix A) and excretion (urine, Appendix A).

### 3.2. CPX Derivatization to Optimize Its Absorption Properties

As discussed previously, CPX is poorly absorbed and rapidly metabolized and excreted via the hepatic route, so it does not attain its full therapeutic potential. Importantly, this is accompanied by acute GI toxicity, as observed in mice (Figure 1A). We hypothesize that the GI toxicity is due to a partial precipitation of the drug in the stomach due to the low pH effect, which provokes a poor absorption and subsequent accumulation in the gut. Such accumulation would increase the effective local exposure of the drug to the tissue above the toxic levels. To overcome this problem, we propose the derivatization of CPX to produce more soluble compounds: generating a phosphate prodrug is one of the common approaches for circumventing poor solubility issues of a parent drug [26]. We expect that, by introducing an ionizable phosphate group to CPX, the phosphate prodrugs will become highly water soluble. More importantly, it is also expected that the phosphate prodrugs will be readily cleaved into CPX by alkaline phosphatase, an enzyme widely distributed in plasma and a variety of tissues [27].

Direct phosphorylation of the hydroxyl group is unstable and leads to immediate hydrolysis (data not shown). Instead, a phosphoryl-oxo-methylene group (*pom*) can be chemically conjugated to the given hydroxyl group to yield a stable entity (Figure 3A, CPX*pom*), as previously described [28]. In principle, CPX*pom* is more prone to be absorbed in the body system, increasing its availability at cellular and subcellular levels. We first investigated the stability and catabolism of CPX*pom* using NMR spectroscopy. As already mentioned, the rationale is that CPX*pom* will be cleaved off by phosphatases, but this reaction does not leave directly to CPX but to a hydroxymethyl derivative (CPX*hm*). This species is chemically unstable, and the hemiacetal is spontaneously hydrolyzed to release CPX and formaldehyde. Remarkably, NMR spectroscopy can detect all the species of the chemical process in serum samples (Figure 3B), with the assignment of protons that unequivocally correspond to the different species under consideration. Of note, we also observed the *de novo* appearance of a chemical shift characteristic for the formic acid proton, most likely a by-product of the oxidation of the released formaldehyde (Figure 3A). In summary, NMR spectroscopy emerges as a powerful methodology to investigate the absorption, distribution, metabolism, and excretion (ADME) of CPX derivatives.

### 3.3. PK and Toxicity Studies of the CPX Derivatives

We then investigated the PK properties of an oral administration of CPX*pom* (gavage) and compared it to the direct administration of the active substance CPX. For each of the formulations, we employed 100 mg/kg, a CPX dose that results in GI toxicity. We also included an intravenous (IV) administration in the experimental design, so the F-value can be estimated. Serum was collected after the administration at 1, 2, 4, 6 (7), 12, 24, and 48 h. The main conclusion of the study is that CPX*pom* does not result in GI toxicity. Consistently, no toxicity was observed in another study that administered the prodrug for 30 days to evaluate its PD properties. As shown in Figure 1B, this administration results in no macroscopic inflammation of the gut and no other associated symptom was observed for the treated mice.

The PK data are summarized in Figure 4 (for serum) and Table 1 (integrated data for serum and urine) and Appendix A. All the quantities refer to CPX*tot* (i.e., the sum of CPX*free* and CPX*glu*). CPX*pom* does not appear in the NMR spectrum so its circulating concentration must be below the detection limit of the technique, while CPX*free* is estimated according to Equation (1), as previously indicated.

The results in serum show that CPX*pom* slightly increases the peak absorption of CPX*tot*: a C_max_ value of 52.87 µg/mL for CPX*tot* when CPX*pom* was administered, as compared to CPX*tot* = 43.4 µg/mL when CPX was administered instead. We attribute these differences to a faster absorption of the prodrug in the intestine, as also suggested by ${\mathrm{AUC}}_{0:24h}^{urine}$ (139 vs. 127 mg/h·mL). Yet, in all cases, CPX*tot* ≃ CPX*glu*, suggesting that the prodrug is quickly converted into the active drug and subsequently into the catabolic by-products, also consistent with the absence of peaks for CPX*pom* in the NMR spectrum.

The shape of the PK profile in serum is also altered when comparing CPX and CPX*pom* administrations. Indeed, the abovementioned C_max_ increase upon CPX*pom* administration is also accompanied by a reduction in the T_1/2_ of almost an hour, underlining an overall change in the PK profile between the drug and the prodrug administration. Again, we hypothesize that CPX administration results in the accumulation of the compound in the gut followed by a more gradual absorption. Altogether, the PK analysis suggests that CPX*pom* may modify some of the ADME properties as compared to the administration of the active principle alone. 

### 3.4. PD Studies of the CPX Derivatives

We first tested the derivatives on different cellular models of the disease, obtained from HEK cells by CRISPR/Cas9 editing [3]. The selected mutations (UROIIIS^C73R^ and UROIIIS^P248Q^) result in destabilized proteins and severe phenotypes [29]. As shown in Figure 5, incubation of the cellular models with both, CPX and CPX*pom*, significantly reduced the levels of the toxic by-product URO I. This is the case for UROIIIS^C73R^ and UROIIIS^P248Q^ (Figure 5C). Docking studies with the CPX*pom* prodrug predict a poor interaction with the binding site of the enzyme, because the N-oxide moiety of the compound (absent in CPX*pom*) is essential to generate stabilizing interactions with the protein. Consistently, equivalent cellular studies with the non-hydrolysable CPX homolog mimosine showed no activity (data not shown), highlighting the relevant role of the N-oxide moiety in the interaction with UROIIIS. Thus, the reported activity for CPX*pom* is attributed to a proper hydrolysis of the compound into the active species. The absence of a lag phase in the kinetic experiment (Figure 5B) indicates that this hydrolysis has to be fast, consistent with the reported literature [30].

We then tested the prodrug in a mouse model of the disease (UROIIIS^P248Q/P248Q^), administering the compound orally by gavage for 30 days. As indicated before, no GI toxicity was observed (Figure 1B), likely due to the absence of accumulation of the active compound in the intestine. Yet, the compound was much less efficient in reducing the circulating levels of toxic porphyrins, and only a 5% reduction in total UROI levels in serum was observed after 30 days, as compared to the 40% reduction observed for an equivalent dose of CPX for the same period. We hypothesize that, among other mechanisms, perhaps UGT enzymes could also glucuronate CPX*hm* at the hydroxymethyl group, thus limiting the capacity of the prodrug to release the active principle.

## 4. Discussion

The currently validated method for CPX quantification relies on drug separation and quantification using high-pressure liquid chromatography (HPLC) [31]. Yet, pure CPX has no optimal properties for HPLC separation, and it requires derivatization, which is accomplished by methylating the weak acidic N-hydroxyl group (pK = 7) of the 1-hydroxy-2(1H)-pyridones with dimethyl sulfate. The resulting 1-methoxypyridones presents a normal chromatographic behavior on silica [31]. Unfortunately, this methodology is not directly applicable to the CPX-related prodrugs and, even for pure CPX, it requires the use of standard compounds for quantification. For that reason, we decided to use NMR spectroscopy, which turned out to be a useful analytical method to investigate CPX catabolism in biofluids (urine and serum). Isolated signals in the spectrum allowed quantification of the different CPX species and the identification of some transient metabolites such as CPX*hm* (Figure 2).

We have then addressed the problem of GI toxicity associated to the oral administration of CPX, an antifungal recently repurposed as a pharmacological chaperone active in CEP. The use of phosphorylated prodrugs (i.e., CPX*pom*) adequately minimized the GI toxicity problem, validating the hypothesis of accumulation of the active principle in the gut due to its poor solubility. The PK studies show an altered PK profile when administering CPX*pom* as compared to CPX, with increased C_max_ and reduced T_1/2_, while maintaining similar (but not identical) clearance rates in urine at 24 h. Considering the scenario where CPX accumulates in the gut region, an uneven absorption between both formulations could explain the differences observed in the PK profiles. 

The PD experiments show disparate results between cellular lines and animal models. While CRISPR/Cas-modified HEK cells show a significant reduction in accumulated toxic porphyrins (Figure 5), an equivalent UROI/COPRO I reduction does not happen in mice after the administration of the prodrug. We attribute this effect to a glucuronidation mechanism targeting CPX*hm* that would limit the amount of CPX*free* that can be released from the prodrug. In any case, these results underline the importance of using animal models in drug discovery, to account for all the complexity provided by the organism.

In summary, the results presented in this study evidence the putative problems for an oral administration of ciclopirox, in line with previous observations [16]. Yet, we also demonstrate that simple modifications in the form of a prodrug to improve solubility may overcome the problem of GI toxicity due to a local accumulation of the drug. Even though the here proposed prodrug does not provide optimal efficacy in its use as a pharmacological chaperone for CEP, other formulations may be able to optimally deliver the drug at high therapeutic efficacy. We are actively pursuing this goal.

## Figures and Tables

**Figure 1 jpm-11-00485-f001:**
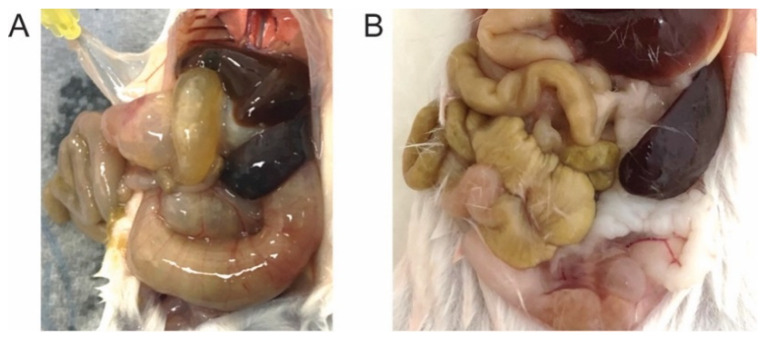
Representative photograph of mice treated with CPX (**A**) and CPX*pom* (**B**). Macroscopic inflammation can be observed for the CPX treatment, evidencing GI toxicity.

**Figure 2 jpm-11-00485-f002:**
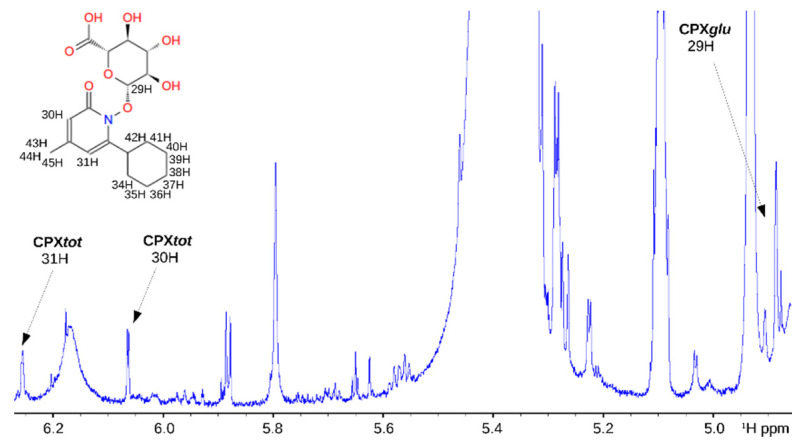
^1^H-1D-NMR spectrum of processed mouse serum resuspended in DMSO-d6 where the assignments of the CPX signals are also shown. CPX*glu* molecule is represented with proton assignment.

**Figure 3 jpm-11-00485-f003:**
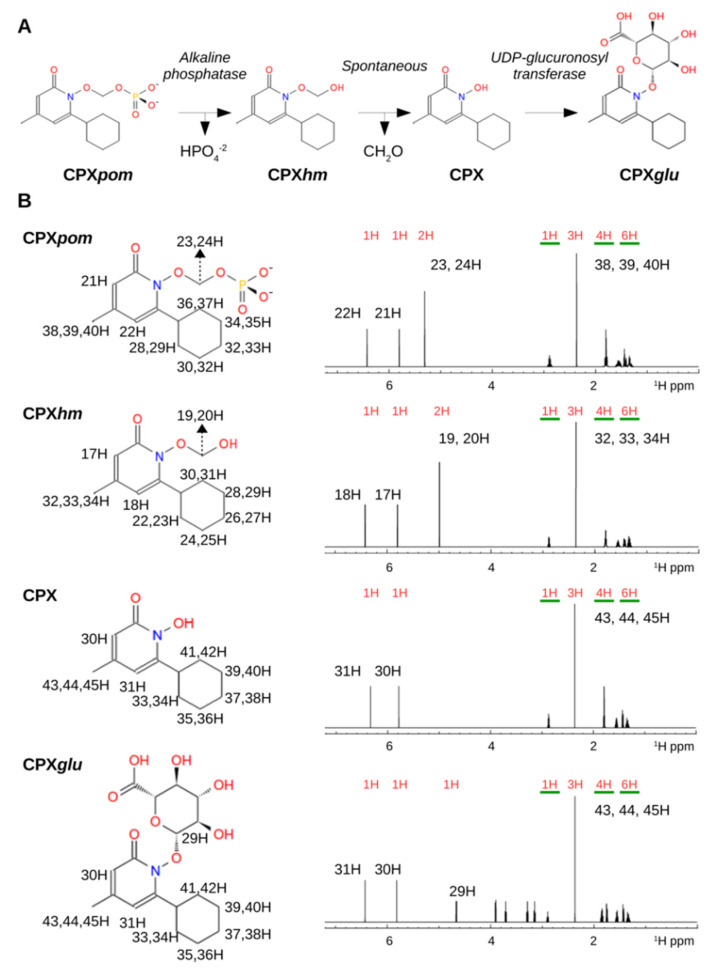
(**A**) CPX*pom* catabolic pathway involving direct phase II biotransformation. (**B**) Theoretical NMR spectra where the expected signals for the different species are shown.

**Figure 4 jpm-11-00485-f004:**
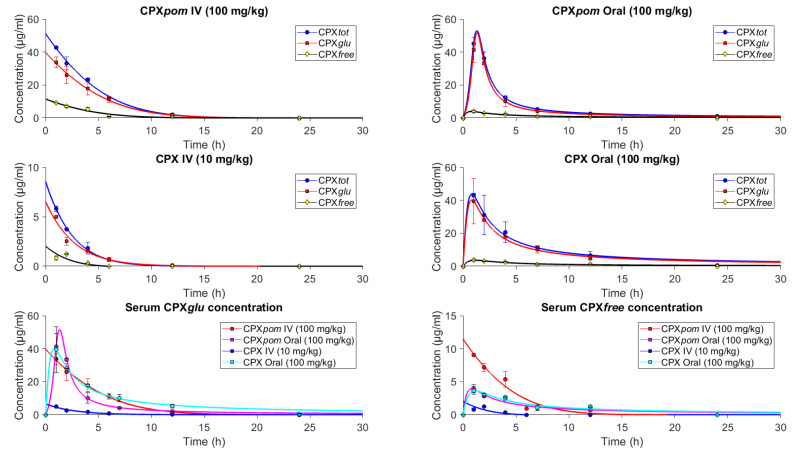
Serum CPX concentrations (μg/mL) as a function of time, as determined by NMR spectroscopy. CPX*tot* is the sum of CPX*free* and CPX*glu*. Mean serum CPX concentrations are shown as squares, whereas the SD values are expressed in bars. Line values correspond to the fitting curve determined for each experiment.

**Figure 5 jpm-11-00485-f005:**
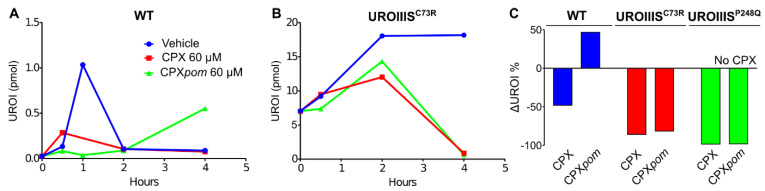
Effect of CPX and derivatives in the UROI concentration as a function of time in cellular models of CEP. (**A**) WT HEK cells, (**B**) HEK cells modified by CRISPR/Cas9 to introduce the mutation C73R in the *UROIIIS* gene, and (**C**) normalized comparative analysis of the URO I concentration in the presence or absence of CPX and CPX*pom* and for two different cellular models of CEP (C73R and P248Q).

**Table 1 jpm-11-00485-t001:** Serum CPX pharmacokinetic parameters in mice following single IV or oral doses of CPX*pom* and CPX. Non-compartmental analysis is assumed for values determination.

Drug		CPX*pom*		CPX	
Via		IV	Oral	IV	Oral
Dose	mg/kg	100	100	10	100
Weight	kg	0.02	0.02	0.02	0.02
DM	mg	2	2	0.2	2
C_max_ ^a^	µg/mL	51.21	52.87	8.56	43.40
T_max_ ^a^	h	0	1.3	0	0.86
T_1/2_ ^a^	h	3.34	2.45	1.75	3.26
AUC_0:12h_ ^a^	µg·(h/mL)	210.99	149.80	24.0	199.20
${\mathrm{AUC}}_{0:24\;h}^{urine\;}$ ^a^	mg·(h/mL)	140.06	139.02	11.80	126.66
F_0:12h_ ^a^	%	71		98	

^a^ Quantities referred to CPXtot. Abbreviations: AUC_0:12h_: area under the curve for serum, integrated at 12 h; ${\mathrm{AUC}}_{0:24\;h}^{urine}$: area under the curve for urine, integrated at 24 h; C_max_: maximum circulating concentration; DM: administered dose; F_0:12h_: (AUC_0:12h_/Dose)_oral_/(AUC_0:12h_/Dose)_IV_; T_1/2_: time at which the concentration becomes half the dose; T_max_: time at C_max_; IV: intravenous administration.

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
