# Peer review of "Improving the Pharmacological Properties of Ciclopirox for Its Use in Congenital Erythropoietic Porphyria"

_jpm, 2021, doi:10.3390/jpm11060485_

Round 1
Reviewer 1 Report
Pers. Med.
Article: "Improving the pharmacological properties of ciclopirox for its 2 use in congenital erythropoietic porphyria".
The titles of the subsections are not separated from their texts; such as in "Materials and Methods": "Compounds and cell Lines". Check across the manuscript.
Spaces lack between many words, e.g. Line 98 "….extraction.Mouse……"; check across the manuscript.
Line 9: Abstract: add "CEP'" after Congenital erythropoietic porphyria.
Line 91 add the common name of the drug after CPX.
Line 109: "CPX derivates extraction". Do you mean CPX metabolites?
Line 119: NMR can be used to identify the pure compound. It is not clear how could you isolated CPX or its metabolites as pure compounds from the serum metabolic extracts.
Line 152: "Results and Discussion": In general, it is known that sulfate-containing compounds are more toxic to the human body, how do you explain this compared to the result of your study
Table 1 must be improved. Title above the table. Give meaning of abbreviations at the table-foot.
There are many abbreviations in the manuscript. It is better to put it together with their meanings in a separate paragraph.
Author Response
The titles of the subsections are not separated from their texts; such as in "Materials and Methods": "Compounds and cell Lines". Check across the manuscript.
It has been corrected, thank you.
Spaces lack between many words, e.g. Line 98 "….extraction.Mouse……"; check across the manuscript.
Checked and corrected, thank you.
Line 9: Abstract: add "CEP'" after Congenital erythropoietic porphyria.
Done.
Line 91 add the common name of the drug after CPX.
Done.
Line 109: "CPX derivates extraction". Do you mean CPX metabolites?
Yes, corrected. Thank you.
Line 119: NMR can be used to identify the pure compound. It is not clear how could you isolated CPX or its metabolites as pure compounds from the serum metabolic extracts.
Sorry, I could not find this sentence on the paper.
Line 152: "Results and Discussion": In general, it is known that sulfate-containing compounds are more toxic to the human body, how do you explain this compared to the result of your study
Sorry, I could not find this sentence on the paper.
Table 1 must be improved. Title above the table. Give meaning of abbreviations at the table-foot.
The title has moved, and the abbreviations are now inserted.
There are many abbreviations in the manuscript. It is better to put it together with their meanings in a separate paragraph.
We have created a dedicated section for the abbreviatures.
Reviewer 2 Report
Improving the pharmacological properties of ciclopirox for its use in congenital erythropoietic porphyria.
This article discusses the repurposing of ciclopirox for the treatment of Gunther's disease. This repurposing is based on the observed chaperone effect of the drug on uroporphyrinogen III synthase; this enzyme is deficient in Gunther's disease and ciclopirox helps to maintain enzyme activity. CPX is a topical drug and the authors suggest that its systemic administration may be associated with high gastrointestinal toxicity related to precipitate formation. The authors optimized a phosphorylated CPX derivative that eliminated gastrointestinal toxicity in mice and maintained activity in cell models; however, activity in the mouse model was limited.
First of all, I wonder if the chosen journal is the most appropriate for this work. Perhaps a more “pre-clinical” journal would be more appropriate (e.g., MDPI Pharmaceuticals or similar). Besides, I would like to suggest to the authors to avoid the use of the present perfect tense in the results section in favor of the past simple tense (e.g., we have investigated-> we investigated). This URL may be of your interest: https://www.nature.com/scitable/topicpage/effective-writing-13815989/
Secondly, the article is interesting and methodologically sound, however, it would be significantly clearer if the results and discussion section would be separated in two different sections.
The article is clear until “PK and toxicity studies for the CPX derivatives”. Table 1 and figure 4 are of great concern. What is AUC and Cmax in the table exactly representing? If total CPX is represented, it should be stated clearly, because it seems as if CPXpom was quantified.
How can a prodrug affect the metabolic characteristics of the active moiety? Line 246 “The results in serum show that CPXpom 246 slightly increases the absorption (Cmax of 52.87 µg/mL for CPXpom as compared to 43.4 247 µg/mL for CPX) but at the expense of a higher metabolic turnover, with a faster excretion” One could argue that, the actual prodrug, is metabolized or excreted and, therefore, the active moiety is obtained to a lesser extent. But how can they explain CPX is being excreted more rapidly if administered as a prodrug? If they mean that the prodrug will be excreted more rapidly, this should be clarified.
It is intriguing that the IV dose of CPXpom was 10 times higher than CPX. However, assuming linear PK, CPX 100 mg/kg IV should lead to a cmax of about 56 ug/ml and an AUC of about 24 ugh/ml. While these values when CPXpom was administered were 51.21 and 210,99, respectively. It seems, therefore, that the elimination or metabolism of CPX is not reduced when administered as a prodrug.
Line 289 “As indicated before, no GI toxicity was observed (Figure 1B), likely due to the absence of accumulation of the active compound in the intestine. Yet, the compound was much less efficient in reducing the circulating levels of toxic porphyrins and only a 5% reduction in total UROI levels in serum was observed after 30 days, as compared to the 40% reduction observed for an equivalent dose of CPX for the same period. We attribute this discrepancy to the accelerated PK profile of the prodrug, which is quickly eliminated in the urine, likely before it can reach the bone marrow and exert its effect” Again, this would be congruent for the prodrug but not for the active moiety. CPX is CPX and its PD should be the same regardless if administered as a prodrug.
Moreover, several UGT enzymes could be responsible for CPX glucuronidation. Could the reduced effectiveness be explained by the glucuronidation of CPXpom or, more likely, of CPXhm at the hydroxymethyl group?
I feel that the results are very interesting but I don't think the authors' explanation of them is quite straightforward. As mentioned, restructuring the manuscript and providing clarity to tables and figures is advisable.
Lastly, in my opinion, a section on future perspectives for this compound is missing. What other alternatives would overcome the low solubility issue but maintain effectiveness?
Author Response
First of all, I wonder if the chosen journal is the most appropriate for this work. Perhaps a more “pre-clinical” journal would be more appropriate (e.g., MDPI Pharmaceuticals or similar). Besides, I would like to suggest to the authors to avoid the use of the present perfect tense in the results section in favor of the past simple tense (e.g., we have investigated-> we investigated). This URL may be of your interest: https://www.nature.com/scitable/topicpage/effective-writing-13815989/
We have eliminated the present perfect tense from the results section.
Secondly, the article is interesting and methodologically sound, however, it would be significantly clearer if the results and discussion section would be separated in two different sections.
We have separated the Results and the discussion sections, as requested.
The article is clear until “PK and toxicity studies for the CPX derivatives”. Table 1 and figure 4 are of great concern. What is AUC and Cmax in the table exactly representing? If total CPX is represented, it should be stated clearly, because it seems as if CPXpom was quantified.
All the quantities refer to CPXtot, understood as the sum of CPXfree and CPXglu. We had no spectroscopic evidence of CPXpom in the spectrum so the circulating concentration must be below the detection limit of the technique. We have clarified this point in the text.
How can a prodrug affect the metabolic characteristics of the active moiety? Line 246 “The results in serum show that CPXpom slightly increases the absorption (Cmax of 52.87 µg/mL for CPXpom as compared to 43.4 247 µg/mL for CPX) but at the expense of a higher metabolic turnover, with a faster excretion” One could argue that, the actual prodrug, is metabolized or excreted and, therefore, the active moiety is obtained to a lesser extent. But how can they explain CPX is being excreted more rapidly if administered as a prodrug? If they mean that the prodrug will be excreted more rapidly, this should be clarified.
We agree with the reviewer that the information provided in the original submission was misleading. We have rewritten the entire section for clarity. In addition, we have clarified the meaning of all the magnitudes under consideration, both in the text and in the caption of Table 1.
It is intriguing that the IV dose of CPXpom was 10 times higher than CPX. However, assuming linear PK, CPX 100 mg/kg IV should lead to a cmax of about 56 ug/ml and an AUC of about 24 ugh/ml. While these values when CPXpom was administered were 51.21 and 210,99, respectively. It seems, therefore, that the elimination or metabolism of CPX is not reduced when administered as a prodrug.
The IV dose of CPX was 10 lower due to the limited solubility of the drug in the vehicle. As for the comparison, there was a mistake in Table 1 and the AUC value for the IV administration of CPX@10mg/Kg was 24 instead of 2.4 (please also notice the area in the given plot of Figure 4). Any case, we totally agree that the elimination of CPX is not reduced when administered as a prodrug, and this has also been clarified in the revised version of the manuscript.
Line 289 “As indicated before, no GI toxicity was observed (Figure 1B), likely due to the absence of accumulation of the active compound in the intestine. Yet, the compound was much less efficient in reducing the circulating levels of toxic porphyrins and only a 5% reduction in total UROI levels in serum was observed after 30 days, as compared to the 40% reduction observed for an equivalent dose of CPX for the same period. We attribute this discrepancy to the accelerated PK profile of the prodrug, which is quickly eliminated in the urine, likely before it can reach the bone marrow and exert its effect” Again, this would be congruent for the prodrug but not for the active moiety. CPX is CPX and its PD should be the same regardless if administered as a prodrug.
Moreover, several UGT enzymes could be responsible for CPX glucuronidation. Could the reduced effectiveness be explained by the glucuronidation of CPXpom or, more likely, of CPXhm at the hydroxymethyl group?
Indeed, we agree!! The experimental evidence clearly showed that CPXpom is much less active than CPX but our reasoning is likely to be ultimately incorrect. Even though we did not notice it, we agree with the reviewer that glucuronidation of CPXhm may be a plausible explanation for the PD data, also explaining why the prodrug is active in cellular lines. We would like to thank the reviewer for this interesting notion, that we have included in the text.
I feel that the results are very interesting but I don't think the authors' explanation of them is quite straightforward. As mentioned, restructuring the manuscript and providing clarity to tables and figures is advisable.
As already discussed, it has been addressed in the revised version of the manuscript.
Lastly, in my opinion, a section on future perspectives for this compound is missing. What other alternatives would overcome the low solubility issue but maintain effectiveness?
We have included a last paragraph in the discussion to address the future perspectives of the project.

Round 2
Reviewer 1 Report
Article: Improving the pharmacological properties of ciclopirox for its use in congenital erythropoietic porphyria
Line 19: "Extraction of CPX-related metabolites"; lines 121-122: "Serum metabolic extracts were resuspended in 480 μl of DMSO-d6 with 1.66 μM of DSS as internal reference (Sodium trimethylsilylpropanesulfonate) and placed into a 5 mm NMR tube"; and line 137: "Chemical shifts for CPXpom, CPXhm, CPX and CPXglu were predicted in silico".
Metabolic extract means that the extract contains many and various metabolites. You must to add a method to isolate the pure compound from the extract o identify it by NMR, due to that the NMR can only identify pure compound and not extract or compounds mixture. It is not clear how could you identified your compounds by NMR if there is no isolation method. OR explain how is NMR able to detect these metabolites as extracts.
Author Response
We agree with the reviewer that "metabolic extraction" is not an accurate term and we have changed it by sample processing. Actually, what we did was to process the serum to remove some of the lipoproteins while retaining the water soluble fraction. That said, we politely disagree with the reviewer in the statement that "NMR can only identify pure compound and not extract or compounds mixture". NMR can identify compounds in a mixture (for instance the entire NMR-based metabolomics field), and this is exactly what we did.
Reviewer 2 Report
Thank you for the restructuring, now the manuscript is clearer.
Author Response
Thank you for the review.